# Mechanisms of Fibroblast Growth Factor 21 Adsorption on Macroion Layers: Molecular Dynamics Modeling and Kinetic Measurements

**DOI:** 10.3390/biom13121709

**Published:** 2023-11-26

**Authors:** Monika Wasilewska, Maria Dąbkowska, Agata Pomorska, Piotr Batys, Bogusław Kowalski, Aneta Michna, Zbigniew Adamczyk

**Affiliations:** 1Jerzy Haber Institute of Catalysis and Surface Chemistry, Polish Academy of Sciences, Niezapominajek 8, 30-239 Krakow, Poland; monika.wasilewska@ikifp.edu.pl (M.W.); agata.pomorska@ikifp.edu.pl (A.P.); piotr.batys@ikifp.edu.pl (P.B.); 2Independent Laboratory of Pharmacokinetic and Clinical Pharmacy, Pomeranian Medical University, Rybacka 1, 70-204 Szczecin, Poland; maria.dabkowska@pum.edu.pl (M.D.); boguslaw.kowalski16@gmail.com (B.K.)

**Keywords:** fibroblast growth factor 21, poly(diallyldimethylammonium chloride), molecular dynamics modeling, adsorption of growth factor, optical waveguide lightmode spectroscopy, quartz crystal microbalance, viability, cytotoxicity, mouse connective tissue fibroblasts, Chinese hamster ovary cell line

## Abstract

Molecular dynamic modeling and various experimental techniques, including multi-angle dynamic light scattering (MADLS), streaming potential, optical waveguide light spectroscopy (OWLS), quartz crystal microbalance with dissipation (QCM), and atomic force microscopy (AFM), were applied to determine the basic physicochemical parameters of fibroblast growth factor 21 in electrolyte solutions. The protein size and shape, cross-section area, dependence of the nominal charge on pH, and isoelectric point of 5.3 were acquired. These data enabled the interpretation of the adsorption kinetics of FGF 21 on bare and macrocation-covered silica investigated by OWLS and QCM. It was confirmed that the protein molecules irreversibly adsorbed on the latter substrate, forming layers with controlled coverage up to 0.8 mg m^−2^, while their adsorption on bare silica was much smaller. The viability of two cell lines, CHO-K1 and L-929, on both bare and macrocation/FGF 21-covered substrates was also determined. It is postulated that the acquired results can serve as useful reference systems for designing complexes that can extend the half-life of FGF 21 in its active state.

## 1. Introduction

Fibroblast growth factors (FGFs) constitute a family of polypeptides and proteins with diverse roles in health and disease. In vertebrates, the group involves 22 members (FGF 1–FGF 23, FGF 15 has not been recognized in this group). They are named based on their ability to stimulate fibroblast proliferation [1,2]. FGFs play crucial roles in the earliest stages of embryonic development and continue to be vital in adulthood, responsible for processes such as regeneration, metabolism, and tissue maintenance. They are expressed in nearly all tissues and involved in regulating the biological responses of angiogenesis; cell adhesion and migration; tissue differentiation; regeneration of damaged skin tissue; formation of blood vessels, muscles, cartilages, bones, teeth, nerves; in wound healing; and metabolism of lipids, sugars, and fats [1]. Furthermore, FGFs play a significant role in tumorigenesis and pulmonary fibrosis and protect against DNA damage induced by oxidants and some environmental toxicants. These 22 members of the FGFs family possess molecular masses from 17 to 34 kDa and share 13–71% amino-acid identity [1,2]. However, it is worth noting that maintaining a precise concentration of FGF in human serum should be strictly defined to effectively regulate a broad spectrum of biological functions.

Fibroblast growth factor 21 (FGF 21), a secreted endocrine hormone, seems to be a promising agent responsible for the regulation of glucose and lipid metabolism. When it is biologically active, it is mainly involved in lowering blood glucose and improving blood lipid profiles [3]. In preclinical animal models, it was confirmed that reductions in blood glucose, insulin, cholesterol, and triglyceride, as well as a decrease in body weight, occurred after administration of FGF 21 [4]. Consequently, this hormone is a promising therapeutic target for metabolic diseases such as type 2 diabetes, obesity, and non-alcoholic steatohepatitis. However, it should be noted that FGF 21 molecules have limited scope of application due to their short half-life (0.5–1.5 h) in vivo and instability in vitro [3].

Despite its immense significance in medicine and biomedical sciences, the physicochemical properties of FGF 21 have not been adequately explored and documented in the scientific literature. Surprisingly, little is known about other essential parameters characterizing the FGF 21 molecule, such as the size and shape, effective charge in electrolyte solutions at various pHs, and, in consequence, about its isoelectric point, which controls the protein solution stability. Also, to our knowledge, no systematic investigation focused on FGF 21’s molecule adsorption kinetics on abiotic surfaces has been reported in the literature. Such measurements, adequately interpreted using the results stemming from theoretical modeling, can furnish valuable information about the above-mentioned parameters, primarily the molecule size, stability, and electrokinetic charge. Given the significant costs of the FGF 21 samples, such parameters are impractical to acquire by classical bulk measurements involving laser Doppler anemometry (LDA).

According to the literature, only the structure of the FGF 21 core was assessed by NMR (nuclear magnetic resonance), whereas molecular dynamics (MD) calculations were applied for the determination of the structure of FGF 21-FGF 19 chimera [5]. The FGF 21 core structure ensemble shows a converged non-canonical β-trefoil conformation that affects the folding of β2-β3 hairpins and, further, overall protein stability. On the other hand, the chimera demonstrates better thermostability without inducing hepatocyte proliferation.

The stability of the highly concentrated FGF 21 (10 g L^−1^) solution over time was determined only in 0.15 M phosphate-buffered saline (PBS) at pH 7.4, containing 30 mM m-cresol at 37 °C by dynamic light scattering (DLS) technique. The finding revealed that FGF 21 rapidly aggregates under these conditions [6]. To reduce FGF 21 aggregation in solutions, the charge of the molecule was increased by the introduction of an additional disulfide bond into the molecule through Leu118Cys and Ala134Cys mutations [6]. The conformational stability of FGF 21 was also examined and confirmed the presence of the three thermal unfolding states [6].

FGF 21, similar to FGF 19 and FGF 23, lacks a heparin-binding domain [7]. It is worth noting that heparin and proteins primarily interact electrostatically due to the presence of sulfonate and carboxyl groups (placed on the heparin chain). Non-electrostatic interactions (hydrogen bonding and hydrophobic interactions) can also have an impact on the stability of the heparin–protein complex [8]. Considering the fact that FGF 21 does not electrostatically interact with negatively charged heparin, one can assume that it should interact with positively charged macroions forming layer. Also, to our knowledge, no systematic investigation focused on FGF 21’s molecule adsorption kinetics on positively charged abiotic surfaces has been reported in the literature.

Poly(diallyldimethylammonium chloride) (PDADMAC) is a strongly positively charged, water-soluble, synthetic macroion possessing a hydrophilic quaternary ammonium group [9]. Applications of biocidal PDADMAC include its use in the design of dental materials [10], water purification [9], and formation of dendronized polymer gelator to trap and immobilize solvent molecules inside [11]. Furthermore, it is frequently used to create anchoring layers in the formation of macroion films [12] and as the positively charged layers in building blocks [13,14]. Accordingly, PDADMAC appears to be a suitable candidate for forming stable complexes with FGF 21 on the surface.

Due to the lack of data, the main aim of this manuscript was the physicochemical characterization of FGF 21 in solution using both theoretical calculations (MD) and an experimental method (multi-angle dynamic light scattering, MADLS). Conducting characterization of FGF 21 molecules in solutions facilitated a more straightforward interpretation of experimental results, including the determination of FGF 21 adsorption and desorption kinetics on/from PDADMAC monolayers. In this part of the study, the following experimental techniques were employed: optical waveguide light spectroscopy (OWLS), gravimetric (quartz crystal microbalance with dissipation, QCM-D), and electrokinetic (streaming potential measurements, SPM), and atomic force microscopy (AFM). It is worth emphasizing that, due to the low detection limit, the applied experimental methods provided reliable results, enabling the determination of the adsorption kinetics and stability of FGF 21 molecules, which have not been previously described in the literature. The impact of the formed PDADMAC-FGF 21 complexes on the viability of two cell lines (Chinese hamster ovary cell line (CHO-K1) and mouse connective tissue fibroblasts (L-929)) was also evaluated.

In addition to their significant importance for basic research, the obtained results may find application in applied studies, enabling the creation of an effective carrier for FGF 21, increasing the stability of the growth factor, and extending its half-life.

## 2. Materials and Methods

### 2.1. Materials

Recombinant human FGF 21 protein (Active), a full-length protein with a molecular mass of 19 kDa, hereafter referred to as FGF 21, was purchased from Abcam (UK) in a form of lyophilized powder. The vial containing FGF 21 was supplemented with an electrolyte solution with ionic strength (*I*) of 0.01 M (NaCl or PBS buffer) at pH 5.8 and 7.4, respectively, prior to the measurements. A solution of FGF 21 with a bulk concentration of 100 mg L^−1^ was obtained. FGF 21 solutions were stored for no longer than 48 h at the temperature of 4 °C.

Poly(diallyldimethylammonium chloride), hereafter referred to as PDADMAC, is a cationic macroion with a molar mass of 101 kg mol^−1^ (number averaged) and 160 kg mol^−1^ (weight averaged) and was purchased from PSS Polymer Standards Service GmbH, Germany. The PDADMAC solutions were prepared by dissolving the appropriate amount of the powder in NaCl solutions of precisely controlled concentration and pH.

Sodium chloride (NaCl), of analytical grade, and phosphate-buffered saline (PBS) were provided by Avantor Performance Materials Poland S.A. and Biomed (Lublin, Poland), respectively, while hydrochloric acid (HCl) was the commercial product of Merck KGaA (Germany) and was used without further purification.

Deionized water, with a resistivity of 18.2 MΩ, was obtained using the Milli-Q Elix & Simplicity 185 purification system from Millipore SAS Molsheim, France.

The temperature of all experiments was maintained at 298 K.

For basal cytotoxicity assessment, two non-cancerous cell lines were used: Chinese hamster ovary epithelial-like cell CHO-K1 (CCL-61™, ATCC^®^, Manassas, VA, USA) and mouse fibroblast subcutaneous connective tissue cells NCTC clone 929 (L-929, CCL-1™, ATCC^®^, Manassas, VA, USA). Both lines are commonly used to test the toxicity of substances.

The complete medium used in experiments was either EMEM (Gibco) with 10% (*v*/*v*) FBS and 1% (*v*/*v*) Pen strep (CHO-K1) or RPMI-1640 (Gibco) with 10% (*v*/*v*) FBS and 1% (*v*/*v*) Pen step (L-929). If not indicated otherwise, all reagents were purchased from Gibco (Thermo Fisher Scientific, Waltham, MA, USA). The cells were maintained at 37 °C in an environment with saturated humidity and 5% CO_2_. The proliferation medium was changed every 2–3 days, and cell passaging was performed once they reached 80% confluence.

### 2.2. Molecular Dynamics Modeling

To provide additional insight, all-atom molecular dynamics (MD) simulations were performed using Gromacs 2022.3 software [15,16]. To describe the protein and counter ions the AMBER03 [17] force field was used, while for water, the explicit TIP3P model was employed [18]. The structure of the FGF 21 core was taken from Ref. [5] (Protein Data Bank ID code 6M6E), while the missing protein parts were modeled using the IntFOLD prediction server [19,20]. Protein and water bonds were constrained by the LINCS [21] and the SETTLE [22] algorithms, respectively, which enabled application of the 2 fs time step. Temperature was maintained at 298 K using the V-rescale thermostat [23] and pressure at 1 bar by using the isotropic Parrinello–Rahman barostat [24]. The coupling constants were 0.5 and 2 ps, respectively. The long-range electrostatic interactions were modelled by the PME method [25], while the Van der Waals interactions were described using the Lennard-Jones potential with a 1.0 nm cut-off, with long-range dispersion corrections. Periodic boundary conditions were applied in all directions. An NPT ensemble run of 200 ns in total duration was preceded by the energy minimization. Two different NaCl concentrations were used, i.e., 0.01 and 0.15 M.

Separately, a multi-protein MD simulation was performed to investigate the aggregation behavior at 0.01 M NaCl. For that, 10 proteins were randomly placed in a 25.6 × 25.6 × 25.6 nm^3^ simulation box, which corresponded to the protein concertation equal to 20 mg/mL. The multi-protein simulations were carried out for 500 ns.

VMD software package (version 1.9.3) was used for visualizations [26]. The protonation states of the residues and the net charge of the protein were calculated using the PROPKA 3.0 algorithm [27,28,29].

### 2.3. Experimental Methods

The stock solutions of FGF 21 at a concentration of 5 mg L^−1^ at pH 7.4, ionic strengths 0.01 M and 0.15 M NaCl, respectively, were used in the multi-angle dynamic light scattering (MADLS).

In the adsorption/desorption experiments, the concentration of FGF 21 was kept constant at 2 mg L^−1^. The pH was adjusted to 5.8 using NaCl and to 7.4 using PBS buffer.

#### 2.3.1. Multi-Angle Dynamic Light Scattering (MADLS)

We used the MADLS technique with the Malvern ZetaSizer Ultra (Malvern Instruments, Malvern, UK) employing ZS XPLORER 3.2.0 software to determine the size distribution of FGF 21 under conditions of 0.15 M or 0.01 M NaCl and pH 7.4. The MADLS approach is a well-established technique for determining the hydrodynamic size distribution of molecules dispersed in solutions, providing reliable data for relatively polydisperse samples. This method improves particle sizing resolution, sensitivity, and accuracy compared to traditional single-angle DLS.

The stock solution of the FGF 21 was diluted to a concentration of 5 mg L^−1^ and passed through a 0.22 μm syringe filter. Kinetic measurements of the hydrodynamic diameters of FGF 21 were taken at various time points: 0 min (immediately after preparing FGF 21 in NaCl at specific ionic strength and pH), 30 min, 1 h, 3 h, 6 h, and 24 h.

#### 2.3.2. Atomic Force Microscopy (AFM)

FGF 21 molecules adsorbed onto PDADMAC-covered silica were imaged using an ambient air AFM NT-MDT Solver BIO device with the SMENA SFC050L scanning head. The concentration of FGF 21 in 0.01 M solution of NaCl was 0.2 mg L^−1^ and the adsorption time was 10 min.

#### 2.3.3. Streaming Potential Measurements (SPM)

The substrates for the adsorption experiments were silicon plates coated with silica SiO_2_ wafers (SIEGERT WAFER GmbH, Aachen, Germany). These wafers were cleaned thoroughly immediately before the experiments, and the cleaning procedures are described in detail in our previous paper.

SPM were conducted in a parallel-plate channel assembled using two SiO_2_ wafers, separated by a Teflon gasket, following the procedure described in previous papers. They were measured in situ using a pair of reversible Ag/AgCl electrodes. Several streaming potential measurements were performed at four different pressure differences that drove the flow through the cell. This allowed for obtaining the slope of the streaming potential vs. hydrostatic pressure difference dependence, which was necessary for calculating the zeta potentials of the bare substrate and PDADMAC monolayer, respectively, in various pHs and ionic strengths. The Smoluchowski equation was applied for the zeta potential calculations [30].

The streaming potential measurements, allowing for determination of zeta potential, consisted of the following steps: (1) calculation of the zeta potential of SiO_2_ wafers, (2) formation of a saturated PDADMAC layer onto silica at pH 5.8, (3) measurement of the streaming potential of the PDADMAC monolayer under various pHs and constant ionic strength, (4) calculation of the zeta potential of the monolayer using the Smoluchowski model.

The initial bulk concentration of PDADMAC was 5 mg L^−1^. The adsorption process occurred inside the streaming potential (SP) cell for a fixed time (20 min) at a volumetric flow rate of 0.02 mL s^−1^. The pH and ionic strength of the PDADMAC solutions were maintained at pH 5.8 and an ionic strength of 0.01 M NaCl. To prevent PDADMAC depletion, all glassware was preconditioned three times with the macroion solutions of the same concentration as those used in the actual experiments.

#### 2.3.4. Optical Waveguide Lightmode Spectroscopy (OWLS)

The kinetics of FGF 21 adsorption from dilute solutions on a PDADMAC layer were determined using the OWLS method, following the procedure described elsewhere [31]. The OWLS 210 instrument (Microvacuum Ltd., Budapest, Hungary) was utilized. The apparatus was equipped with a laminar slit shear flow cell comprising a silica-coated waveguide (OW2400, Microvacuum). The adsorbing substrates were planar optical waveguides made of a glass substrate (refractive index 1.526) covered by a 170 nm thick film of Si_0.78_Ti_0.22_O_2_ with a refractive index 1.8. A grating embossed in the substrate enabled the coupling of light into the waveguide layer. The sensor surface was further coated with an additional 10 nm layer of pure SiO_2_ [31].

In situ OWLS measurement involved several steps. Initially, a pure electrolyte with a defined ionic strength and pH was passed through the cell (the flow velocity of the electrolyte was set to 2.5 × 10^−3^ cm^3^ s^−1^) to condition the sensor surface and establish a stable baseline. During the initial adsorption step, molecules from the PDADMAC solution, flowing over the sensor at the same velocity as the pure electrolyte, were adsorbed onto the sensor. The stability of the adsorbed layer was checked by flushing the OWLS cell with pure electrolyte. In the subsequent step, FGF 21 was introduced into the cell, and its molecules were adsorbed onto the adsorbed PDADMAC layer. The stability of the bilayer formed as such was also determined.

It is worth noting that upon adsorption onto the sensor’s surface, the effective refractive indices shifted to higher values, enabling the real-time monitoring of the macroion adsorption kinetics.

#### 2.3.5. Gravimetric Measurements (QCM-D)

Silica-coated AT-cut quartz crystals (Si/SiO_2_) with a 5 MHz fundamental frequency were purchased from Qsense/Biolin Scientific for gravimetric measurements with the Quartz Crystal Microbalance with Dissipation (QCM-D, Qsense E1 flow module) technique. These crystals were submerged in 30 min-old, dissolved piranha solution (1:1:1-H_2_SO_4_:H_2_O_2_:H_2_O) for 2 min, rinsed with 80 °C ultrapure water, and dried out in a stream of pure nitrogen prior to each measurement.

The kinetics of the PDADMAC/FGF 21 bilayer formation on SiO_2_-coated sensors were determined using a Q-Sense E1 system (QSense, Gothenburg, Sweden).

Firstly, a stable baseline in pure electrolyte (NaCl, *I* = 0.01 M, pH 5.8 or pH 7.4) was attained in the QCM-D cell for a defined liquid flow of 1.33 × 10^−3^ cm^3^ s^−1^. Next, a PDADMAC solution with a bulk concentration of 5 mg L^−1^ (in the same electrolyte) was introduced into the cell and the macrocation adsorption was monitored for 30 min. Subsequently, the adsorbed layer was flushed by the pure electrolyte (at the earlier fixed flow rate) and the desorption process was monitored for 30 min. The deposition process of FGF 21 was carried out in the same conditions (from protein solution with the bulk concentration of 2 mg L^−1^) for 20 min, followed by a rinsing step (30 min). The measurements were carried out at 298 K.

The QCM coverage of the protein was calculated from the Sauerbrey equation [32]
(1)$$\Gamma_{Q}=-C_{Q}\frac{\Delta f}{n_{o}}$$
where: $\Gamma_{Q}$ is the protein coverage, *C_Q_* is the Sauerbrey constant equal to 0.177 mg m^−2^ Hz^−1^ [33] for the 5 MHz AT-cut quartz sensor, Δf is the frequency change, and *n_o_* is the overtone number (the third to eleventh overtones were used).

### 2.4. Cell Viability Study

#### 2.4.1. Preparation of Coatings for In Vitro Studies

The PDADMAC macroion was employed as a carrier for FGF 21 molecules. Before adsorption, macroion solutions with a bulk concentration of 250 mg L^−1^ at 0.01 or 0.15 M and pH levels of 4.0, 5.8, or 7.4 were diluted to a concentration of 5 mg L^−1^ to obtain a solution with a strictly defined pH and *I*. In the next step, this diluted PDADMAC solution was adsorbed on a 96-well plate (Thermo Fisher Scientific, Waltham, MA, USA) for 20 min. After adsorption, the wells were rinsed thrice with NaCl at pH levels of 4.0, 5.8, or 7.4.

Stock solutions of FGF 21 were prepared at concentrations of 100 mg L^−1^ in PBS with a pH of 7.4 ± 0.2 and an ionic strength of 0.15 M, shortly before being used for adsorption onto the PDADMAC-coated plate. The adsorption of FGF 21 onto the PDADMAC was carried out at room temperature for 40 min. Subsequently, the FGF 21/PDADMAC bilayer underwent three rinses with 0.01/0.15 M NaCl at pH levels of 4.0/5.8/7.4. After this preparation step, CHO-K1/L-929 cells were seeded onto these surfaces.

The experimental procedure was conducted under controlled conditions in an Alpina chamber with laminar airflow and HEPA filters. These conditions were maintained to ensure the accuracy and reliability of the in vitro data obtained.

#### 2.4.2. Cell Viability

For the cell viability assessment, CHO-K1/L-929 cells were seeded at a density of 10^4^ cells per well in a 96-well plate in the following experimental conditions: control (no PDADMAC layer, no FGF 21), PDADMAC, FGF 21, PDADMAC/FGF 21 layer. During the adherence, cells were incubated for 24 h in a culture medium, and the viability of CHO-K1/L-929 cells was assessed using the Alamar Blue^®^ reagent assay (Thermo Fisher Scientific, Waltham, MA, USA) following the manufacturer’s instruction.

Fluorescence readings were obtained using a Varioskan™ LUX multimode microplate reader (Thermo Fisher Scientific, Waltham, MA, USA). The data were analyzed using SkanIt™ Software (version 7.0.0.50) for Microplate Readers ver. 5.0.

#### 2.4.3. Statistical Analysis

The data in this study are presented as means ± standard deviations (SD) and are based on a minimum of three independent experiments. One-way ANOVA and Dunnett’s multiple comparisons test, performed using GraphPad Prism (version 10.0.1 for Windows, GraphPad Software, San Diego, CA, USA, accessed date: 8 February 2023 www.graphpad.com), were used to perform statistical analysis between each study group in comparison to the control. For analysis comparing experimental groups, a one-way ANOVA was utilized. Statistical significance was determined as follows: * *p* < 0.05, ** *p* < 0.01, *** *p* < 0.001, **** *p* < 0.0001.

## 3. Theoretical Modeling Results—Molecule Structure and Charge

Considering that the FGF 21 molar mass derived from the primary structure, denoted by *M_w_*, is equal to 19,400 g mol^−1^ (Da) the volume of the single molecule (monomer) can be calculated from the dependence
(2)$$v_{1}=M_{w}/(N_{Av}\rho_{p})$$
where $N_{Av}$ is the Avogadro number equal to 6.02 × 10^23^ and $\rho_{p}$ is the protein density.

Taking the typical density pertinent to globular proteins equal to 1.35 g cm^−3^ [34], one obtains from Equation (2) that *v*_1_ = 23.9 nm^3^. Consequently, the diameter of the equivalent sphere can be calculated from the equation
(3)$$d_{1}={\left(\frac{6v_{1}}{\pi }\right)}^{1/3}$$

One obtains from Equation (3) that $d_{1}=$ 3.57 nm, which is a reasonable estimate of the real size of the FGF 21 molecule. Interestingly, as discussed in Ref. [35], the hydrodynamic diameter of the aggregate composed of two spherical molecules is 1.39 times larger than the hydrodynamic diameter of a single particle. This gives 5.0 nm as a plausible value for the hydrodynamic diameter of the FGF 21 molecule dimer.

The real shape of the molecule was derived from the MD modeling performed using the algorithm described above. The snapshots shown in Figure 1 for various molecule orientations and the NaCl concentrations of 0.01 and 0.15 M confirm that the FGF 21 molecule assumed a globular shape with an approximate maximum and minimum dimension of 4.5 and 3.5 nm. The average cross-section area of the molecule ($S_{g}$) was determined to be 17 nm^2^ from the modeling.

Using this value, one can calculate the maximum mass coverage of the molecules from the formula
(4)$$\Gamma_{mx}=\frac{M_{w}}{N_{Av}}\frac{\theta_{mx}}{S_{g}}$$
where $\theta_{mx}$ = 0.547 is the maximum jamming coverage for hard (non-interacting) molecules of spherical shape [36]. In this way, one obtains 1.0 mg m^−2^ as the maximum coverage of the FGF 21 molecules. However, at lower ionic strength, the coverage is expected to be smaller because of the appearance of lateral electrostatic interactions among the molecules, usually being of a repulsive type.

To complete the physicochemical characteristics, the fixed (nominal) charge of the FGF 21 molecule was calculated using the PROPKA algorithm [27,28,29]. The results are shown in Figure 2 with the dependence of the charge expressed as the number of elementary positive charges. As can be seen, the FGF 21 molecule charge is equal to 8 e at pH 4 and then monotonically decreases, attaining 0 at pH 5.3, and −9 e at pH 10. The exact numerical data were fitted for the pH range from 4 to 8 by the following function, with R^2^ > 0.99,
*Q*_p_ (pH) = *C*_0_ − *C*_1_ pH + pH^2^(5)
where: *C*_0_ = 55.9; *C*_1_ = 15.9

Additionally, the feasibility of FGF 21 molecule aggregation was theoretically investigated using the MD modeling. In Figure 3, snapshots of the molecule solution derived from the MD modeling at the beginning and after 500 ns are shown. The tendency towards dimer formation (marked in red) is clearly visible.

## 4. Experimental Results

### 4.1. FGF 21 Molecule and the Substrate Characteristics

The dependence of the diffusion coefficient of FGF 21 molecules in solutions of different NaCl concentrations was determined as described above by MADLS. The presence of one (by number) and three peaks (by intensity) (see Supporting Information), respectively, was confirmed in these measurements. Using these data, the molecule hydrodynamic diameter, *d*_H_, was calculated from the Stokes–Einstein relationship:(6)$$d_{H}=\frac{kT}{3\pi \eta D}$$
where *k* is the Boltzmann constant, *T* is the absolute temperature, *η* is the dynamic viscosity of the electrolyte, and *D* is the diffusion coefficient derived from MADLS.

It is worth mentioning that compared to the diffusion coefficient, the hydrodynamic diameter is a more universal parameter because it is independent of the temperature and solvent viscosity.

The results, where *d*_H_ was determined based on the size by number, obtained at pH 7.4 with 0.15 and 0.01 M NaCl, are shown in Figure 4. They indicate that the hydrodynamic diameters corresponding to the first two peaks were equal to 3.7 and 5.4 nm. Interestingly, the first value matches the theoretically predicted hydrodynamic diameter of the equivalent sphere equal to 3.6 nm, whereas the second value is close to a FGF 21 molecule dimer diameter predicted from the MD modeling whose hydrodynamic diameter is equal to 5.1 nm. These values remained constant over 1400 min, indicating that the FGF 21 solution was stable. It should be mentioned, however, that in FGF 21 solutions at lower pHs the presence of a small number of aggregates characterized by considerably larger hydrodynamic diameter was also detected.

In order to quantitatively interpret the FGF 21 molecule adsorption kinetics on bare and PDADMAC-modified silica sensors, their zeta potential is needed. These parameters were determined as a function of pH for different ionic NaCl concentrations using the sensitive SPM. The results shown in Figure 5 indicate that the zeta potential of the bare silica was negative for the pH range from 3 to 12 and was equal to −25 mV and −65 mV at pH 4 and 7.4, respectively, for an NaCl concentration of 0.01 M. The zeta potential of silica at an ionic strength of 0.1 M was slightly higher and amounted to −20 mV (pH 4) and −39 mV (pH 7.4), respectively. Additionally, in Figure 5, the dependence of the zeta potential of PDADMAC on pH determined in bulk is presented. As can be seen, the zeta potential of PDADMAC was practically independent of pH and remained around 70 mV. In contrast, the zeta potential of the PDADMAC layer on silica was positive for the pH range from 4 to 7.4 and was equal to 20 mV. These values are significantly lower than those obtained in the bulk measurements. Accordingly, the obtained results confirm that PDADMAC molecules take on elongated rod-like shapes during adsorption and monolayer formation. The extended shape of PDADMAC molecules during adsorption on a flat substrate (mica) and their mostly side-on conformation during adsorption was established in our previous paper [37]. Accordingly, the PDADMAC coverage was low and equaled ~0.1.

### 4.2. Adsorption Kinetics of FGF 21 Molecules—OWLS and QCM Measurements

Thorough OWLS and QCM kinetic measurements were performed to determine the optimum conditions for the formation of stable FGF 21 layers at the silica substrates with well-controlled coverage. The combination of these two techniques is particularly advantageous because the former yields precise data but is rather time-consuming and prone to temperature fluctuation disturbances. In contrast, the QCM is more precise and time-efficient but the interpretation of primary signals (frequency and dissipation shifts) requires independent measurements carried out by other techniques

The adsorption kinetics of FGF 21 on bare silica determined by OWLS at pH 4 and 7.4 are presented in Figure 6 and Figure 7, respectively. It is worth mentioning that this technique also enabled us to determine the molecule desorption kinetics, which were realized by switching to the pure electrolyte flow; this is shown by arrows in Figure 6 and Figure 7. As can be seen, the kinetics were characterized by a rapid increase in the protein coverage, which attained the peak values of 0.35–0.4 mg m^−2^ (for 0.01 M NaCl) and 0.25 mg m^−2^ (for 0.15 M NaCl at pH 7.4). After initiating the desorption runs, the protein coverage decreased and attained a minimum value of 0.2 mg m^−2^ at pH 7.4 for 0.01 and 0.15 M NaCl, respectively. Given that this limiting coverage is much smaller than the theoretically predicted value of 1.0 mg m^−2^, one can conclude that FGF 21 adsorption on bare silica is rather inefficient. This behavior can be attributed to the fact that the silica substrate exhibits a negative zeta potential at both pHs (see Figure 4), prohibiting a stronger binding of FGF 21 molecules, which bear only a small positive (at pH 4) or negative charge (at pH 7.4) (Figure 2).

This conclusion is confirmed by the OWLS kinetic measurements carried out for the PDADMAC-covered silica shown in Figure 8. One can observe that, in this case, the adsorption of FGF 21 is significantly more pronounced because the peak value attained ca. 1 mg m^−2^ (relative to the PDADMAC adsorption of ca. 0.4 mg m^−2^), which decreased after initiating the desorption run to ca 0.8 mg m^−2^. This is close to the maximum value theoretically predicted for a saturated protein layer.

In order to derive additional information about the mechanism of FGF 21 molecule adsorption on PDADMAC, a short time part of the kinetic run shown in Figure 8 was analyzed in more detail and compared with analogous kinetics previously acquired for human serum albumin (HSA) [31]. The results shown in Figure 9 indicate that FGF 21 adsorption up to the time of 450 s remains linear with respect to the time, which confirms that it was controlled by bulk transport and that the protein aggregation during adsorption was minimal [31]. This conclusion is confirmed by the absence of larger-size aggregates in the FGF 21 layer on PDADMAC imaged by AFM (see Figure 9b). Interestingly, in this image the presence of long PDADMAC chains decorated by the protein molecules is clearly visible. Such elongated macroion structures were theoretically predicted in Ref. [37].

In addition, using the kinetic data shown in Figure 9, one can determine the hydrodynamic diameter of the protein molecules under dynamic flow conditions, whereas the MADLS measurements discussed above only yield results obtained under static conditions. The procedure previously applied for the chitosan molecules [40] exploits the fact that the adsorption kinetics of the protein molecules in the OWLS cell under the low coverage regime are described by the formula SI [40]
(7)$$\Gamma =k_{c}^{0}D^{2/3}Q_{}^{1/3}c_{b}t$$
where Γ is the protein mass coverage, usually expressed in mg m^−2^, $k_{c}^{0}$ is the mass transfer rate constant, *D* is the molecule diffusion coefficient, *Q* is the volumetric flow rate, *c*_b_ is the bulk concentration, and *t* is adsorption time.

Because of the linearity of Equation (7) the slope of the dependence of the coverage on the adsorption time $\frac{\Delta \Gamma }{\Delta t}\;$, denoted by *s_l_*, is given by
(8)$$\frac{\Delta \Gamma }{\Delta t}=s_{l}=k_{c}^{0}D^{2/3}Q^{1/3}c_{b}=k_{c}^{0}{\left(\frac{kT}{3\pi \eta d_{H}}\right)}^{\frac{2}{3}}Q^{1/3}c_{b}$$
where *d*_H_ is the hydrodynamic diameter of the molecule to be determined.

Equation (8) can be rearranged to the form directly yielding the hydrodynamic diameter of the protein molecules
(9)$$d_{H}=\frac{kT}{3\pi \eta }\;{\left(\frac{k_{c}^{0}Q_{}^{1/3}c_{b}}{s_{l}}\right)}^{\frac{3}{2}}$$

In order to avoid the calculation of the mass transfer constant $k_{c}^{0}$ one can apply a more straightforward procedure based on the calibration method using a protein of a known hydrodynamic diameter *d*_H0_. In this case, Equation (9) can be formulated as
(10)$$d_{H}=d_{H0}\;{\left(\frac{s_{l}{}_{0}^{}Q_{}^{1/3}c_{b}}{s_{l}Q_{0}^{1/3}c_{b0}}\right)}^{\frac{3}{2}}$$
where $s_{l}{}_{0}^{}$ is the slope of the linear adsorption kinetic range for the reference protein, $Q_{0}^{1/3}$ is the flow rate, and $c_{b0}$ is the bulk concentration of the reference protein.

In this work, we used HSA as a reference protein, characterized by the hydrodynamic diameter of 7.5 nm [31]. The adsorption kinetics of HSA molecules under the linear regime are compared with the FGF 21 adsorption kinetics in Figure 9a. The slope of this dependence $s_{l}{}_{0}^{}$ is equal to 5.5 × 10^−3^ L m^−2^ s^−1^ (flow rate 2.5 × 10^−3^ cm^3^ s^−1^), *c*_b_=5 mg L^−1^ whereas the FGF 21 slope $s_{l}$ is equal to 2.2 × 10^−3^ L m^−2^ s^−1^ (flow rate 1.1 × 10^−3^ cm^3^ s^−1^), *c*_b_=2 mg L^−1^. Considering these data, one can calculate from Equation (10) that the hydrodynamic diameter of the FGF 21 molecules was equal to 5.0 nm, which corresponds to the hydrodynamic diameter of the dimers derived from MADLS.

The adsorption kinetics of FGF 21 molecules on PDADMAC-covered silica sensor was also investigated by QCM. A representative adsorption/desorption kinetic run acquired for FGF 21 bulk concentrations of 2 mg L^−1^, an NaCl concentration of 0.01 M, and pH of 7.4 is shown in Figure 10. As previously for OWLS, the protein coverage rapidly increased in a linear way, attaining plateau values equal to 1.7 and 1.9 mg m^−2^ (for the 11th and 3rd overtone, respectively) after a time of 10 min. Upon initiation of the desorption run, the FGF 21 coverage decreased to 1.5 and to 1.7 mg m^−2^ (for the 11th and 3rd overtone, respectively). However, it should be pointed out that the protein coverage derived from the QCM measurements does not correspond to the accurate physical (dry) coverage because of the appearance of a hydrodynamic perturbing force on particles oscillating with the sensor [31]. Therefore, the real (dry) protein coverage Γ is given by the expression
(11)$$\Gamma_{}=\Gamma_{Q}(1-H)$$
where $\Gamma_{Q}$ is the protein coverage derived from QCM measurements and *H* is the dimensionless correction function accounting for the hydrodynamic interactions previously determined to be equal to 0.5 for globular proteins such as HSA [31].

Using this value one can calculate that the real FGF 21 coverage of irreversibly bound molecules is equal to 0.8 mg m^−2^ (average value calculated from the overtones 11th and 3rd), which matches the OWLS value.

One can conclude that the experimental results derived from OWLS and QCM confirmed that one could irreversibly adsorb the FGF 21 molecule layers on PDADMAC-covered sensors, characterized by a well-controlled coverage up to 0.8 mg m^−2^. Such layers can be effectively used for the robust immobilization of larger biomolecules, for example, living cells.

### 4.3. Cell (L-929) Viability

After the thorough physicochemical characterization of the PDADMAC/FGF 21 coatings, they were applied to test the viability of two non-cancerous cell lines: L-929 and CHO-K1 (Figure 11 and Appendix A, respectively).

When the cells were cultured on the PDADMAC layer (0.01 M NaCl pH 5.8), a significant decrease in viability (88.9 ± 2.9%) was observed for the L-929 cells. Generally, cell viability was lower when cultured on the PDADMAC layer; nevertheless, we did not observe differences in cell viability between the control and cells cultured on the PDADMAC layers formulated with an ionic strength of 0.15 M. Our results agree with the studies described by Fischer at al. [41]. The authors found that the impact of the PDADMAC alone on cytotoxicity in L-929 mouse fibroblasts was lower than for other macrocations, such as poly(ethylenimine) and poly(l-lysine), which are commonly applied as anchoring layers. Furthermore, it was found that the magnitude of the cytotoxic effects of all macrocations are time- and concentration-dependent. Also, molecular weight as well as charge density have an impact on the interaction with the cell membranes and, consequently, cell damage.

It is worth pointing out that no toxicity in the L-929 cell line was determined after 24 h of incubation with FGF 21 alone. Our results agree with the studies of Zhu et al., who found that FGF 21 significantly enhanced the viability of human umbilical-vein endothelial cells [42].

Adsorption of FGF 21 onto the PDADMAC layer (where the coating was formed by adsorption from the macroion solution of 0.01 M NaCl at pH 5.8/7.4) reduced the viability of L-929 cells; however, the viability was higher (than for PDADMAC layer alone) and was equal to 91.5 ± 6.5%. Interestingly, results from the AlamarBlue assay indicated that neither PDADMAC nor the PDADMAC/FGF 21 coatings, obtained from 0.15 M NaCl and pH 4.0, led to any significant deterioration in the studied cell line. One potential explanation is that the formulation of the layer at a specific ionic strength and pH may exert a biological effect on cells, making the system more biocompatible.

## 5. Conclusions

A basic physicochemical characterization of FGF 21 molecules in electrolyte solutions was acquired for the first time applying molecular dynamics modeling and multi-angle dynamic light scattering. The globular molecular shape at different ionic strengths was confirmed, and their dimensions and the cross-section area were established. Additionally, the dependence of the nominal charge of the protein on pH was calculated, yielding an isoelectric point of 5.3. The tendency of FGF 21 to form dimers in bulk was also confirmed by MD and MADLS.

The adsorption kinetics experiments carried out under flow conditions revealed that FGF 21 irreversibly adsorbed onto PDADMAC-covered silica., forming layers with a controlled coverage up to 0.8 mg m^−2^, whereas their adsorption on bare silica was much smaller.

It was also observed that FGF 21 is not toxic to either of the examined cell lines (L-929 and CHO-K1), while PDADMAC alone reduces cell viability. On the other hand, the PDADMAC-FGF 21 complex was less toxic to the cells than PDADMAC alone. Therefore, the complex formation appears to mitigate the cytotoxic properties of PDADMAC and may have a positive impact on cell adhesion, proliferation, and overall cell health.

One can expect that the formation of stable FGF 21-biocompatible macrocation complexes will extend the half-life of the growth factor in its active state. Furthermore, when the macrocation is appropriately chosen, it can also affect cell viability, especially that of fibroblasts, their adhesion, and proliferation.

## Figures and Tables

**Figure 1 biomolecules-13-01709-f001:**
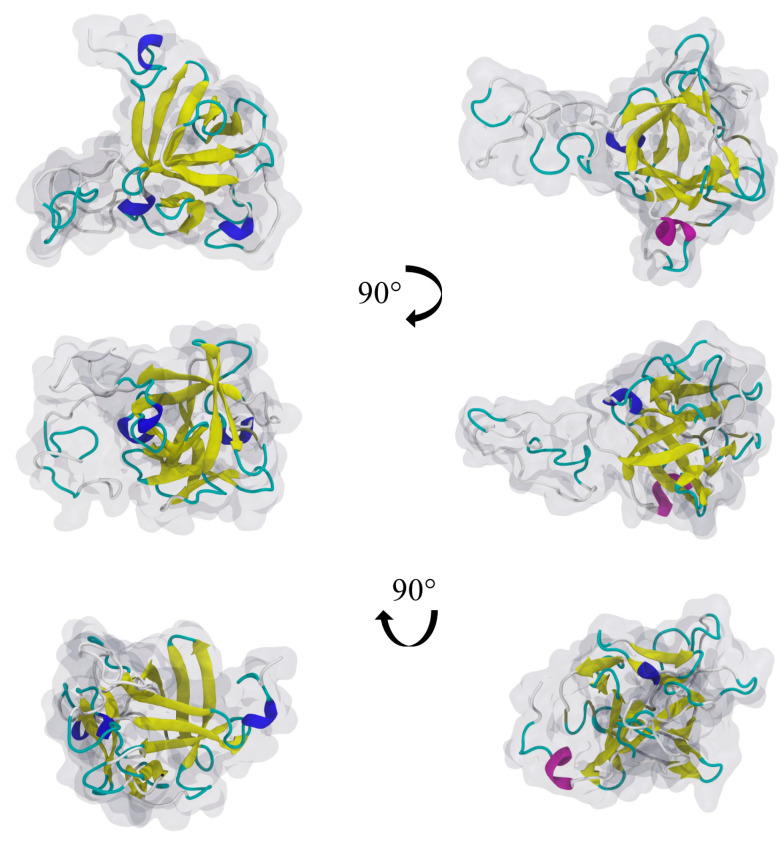
FGF 21 molecules’ shape and secondary structure, for three different orientations, after 300 ns MD simulations in NaCl solutions of different concentrations: 0.01 M (on the **left**) and 0.15 M (on the **right**). Semi-transparent gray shapes correspond to van der Waals radii.

**Figure 2 biomolecules-13-01709-f002:**
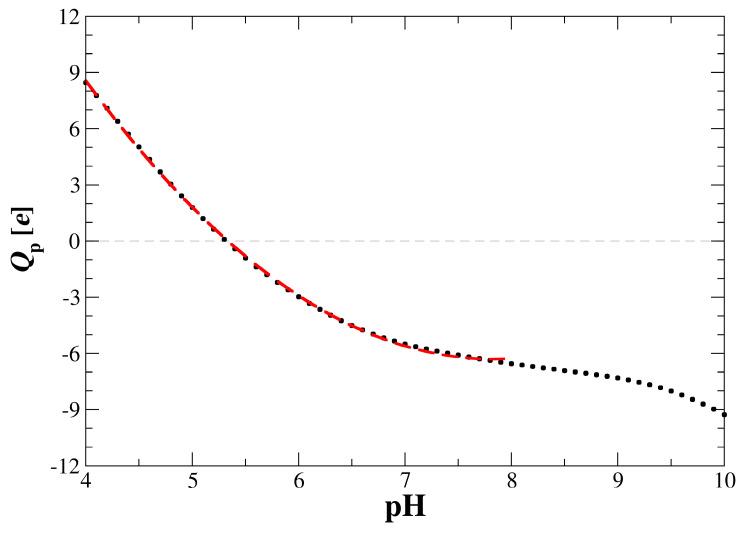
The nominal charge (*Q*_p_) of the FGF 21 molecule versus pH calculated using the PROPKA 3.0 algorithm (the dotted line). The dashed red line represents the charge calculated from the polynomial fitting function described by Equation (5).

**Figure 3 biomolecules-13-01709-f003:**
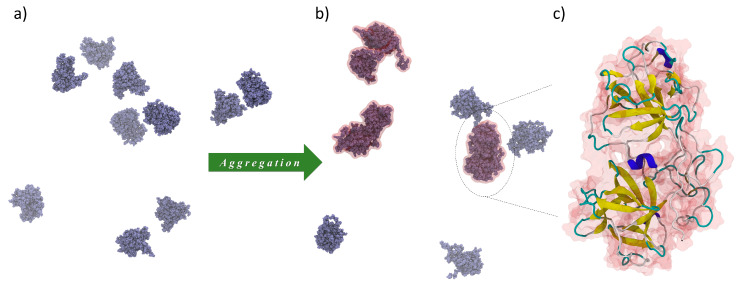
The snapshots of the FGF 21 molecule solution (**a**) at the beginning and (**b**) after the time of 500 ns, derived from MD modeling, show the presence of monomers as well as dimers (marked by red). The zoom of a selected dimer is presented in (**c**).

**Figure 4 biomolecules-13-01709-f004:**
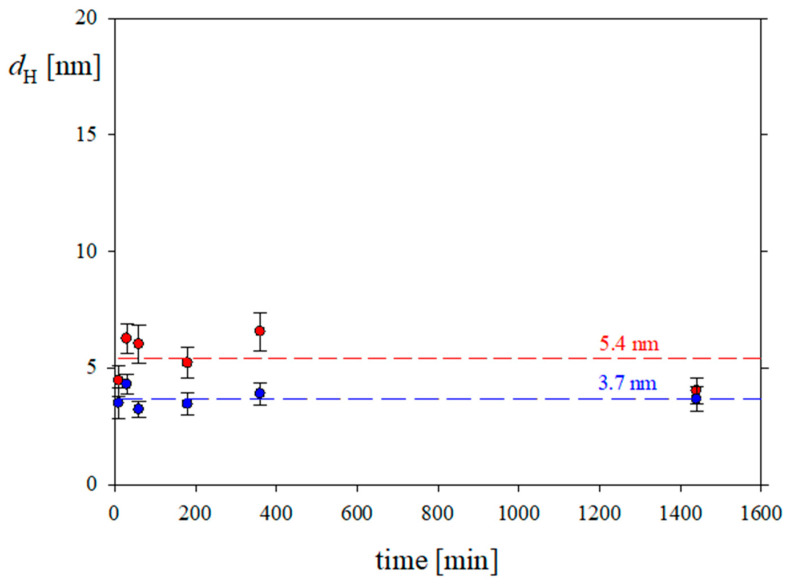
The dependences of the hydrodynamic diameter (*d*_H_) of the FGF 21 molecule on time calculated from the Stokes–Einstein relationship formula at pH 7.4 and ionic strengths of 0.01 M (•) and 0.15 M (•). The straight dashed lines are a guide to the eyes.

**Figure 5 biomolecules-13-01709-f005:**
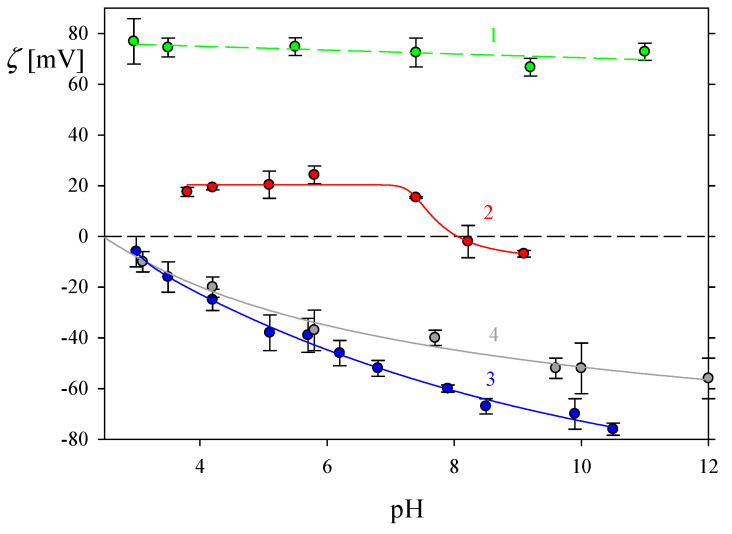
Dependencies of the zeta potential of various substrates on pH derived from the SPM in NaCl solutions of different concentrations. 1. Bulk zeta potential of PDADMAC on pH in 0.01 M NaCl (•). 2. Silica covered by a saturated PDADMAC layer, 0.01 M NaCl (•). 3. Bare silica, 0.01 M NaCl (results from Ref. [38]) (•). 4. Bare silica, 0.1 M NaCl (results from Ref. [39]) (•).

**Figure 6 biomolecules-13-01709-f006:**
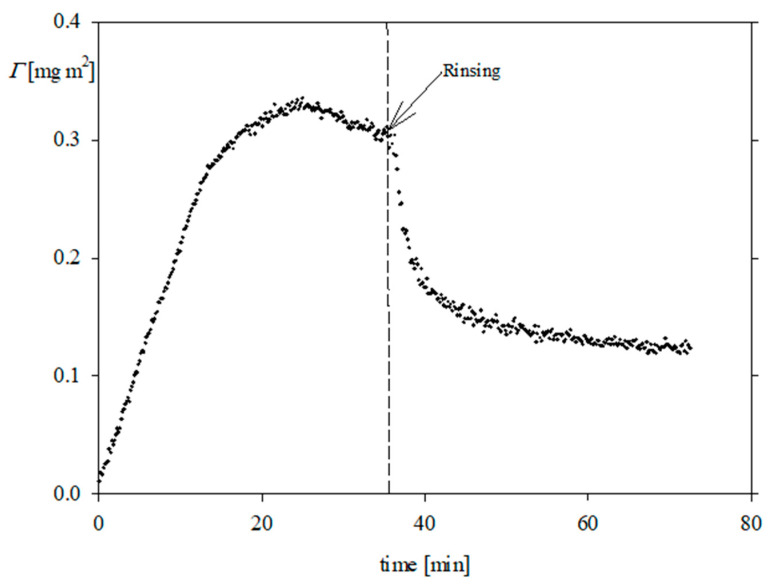
Kinetics of FGF 21 (•) molecule adsorption on bare silica determined by OWLS; the arrow shows the beginning of the desorption run (rinsing step); bulk protein concentration 2 mg L^−1^, flow rate 1.1 × 10^−3^ cm^3^ s^−1^, pH 4.0, 0.01 M NaCl.

**Figure 7 biomolecules-13-01709-f007:**
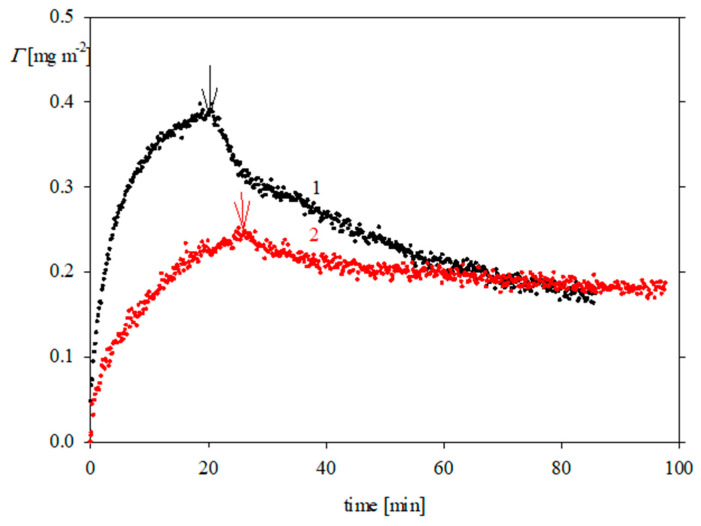
Kinetics of FGF 21 molecule adsorption on bare silica determined by OWLS; the arrows show the beginning of the desorption runs; bulk protein concentration 2 mg L^−1^, flow rate 1.1 × 10^−3^ cm^3^ s^−1^, pH 7.4, (1) 0.01 M NaCl; (2) 0.15 M NaCl.

**Figure 8 biomolecules-13-01709-f008:**
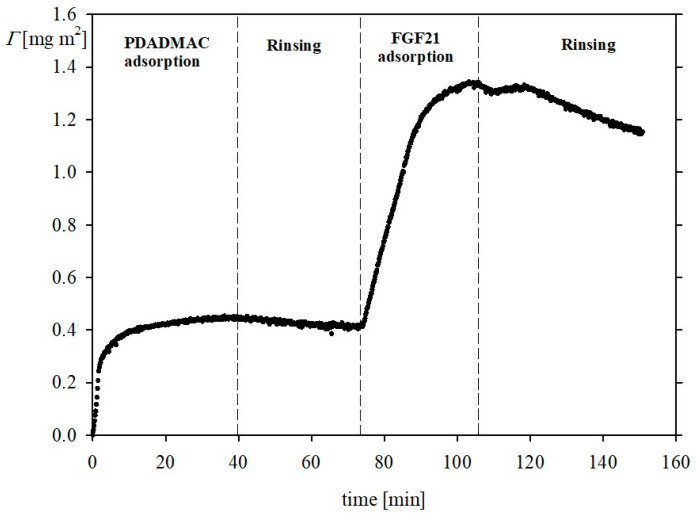
Kinetics of FGF 21 molecule adsorption on the silica/PDADMAC substrate determined by OWLS; the dashed lines show the beginning of the desorption runs (rinsing step); bulk protein concentration 2 mg L^−1^, flow rate 1.1 × 10^−3^ cm^3^ s^−1^, pH 7.4, 0.01 M NaCl.

**Figure 9 biomolecules-13-01709-f009:**
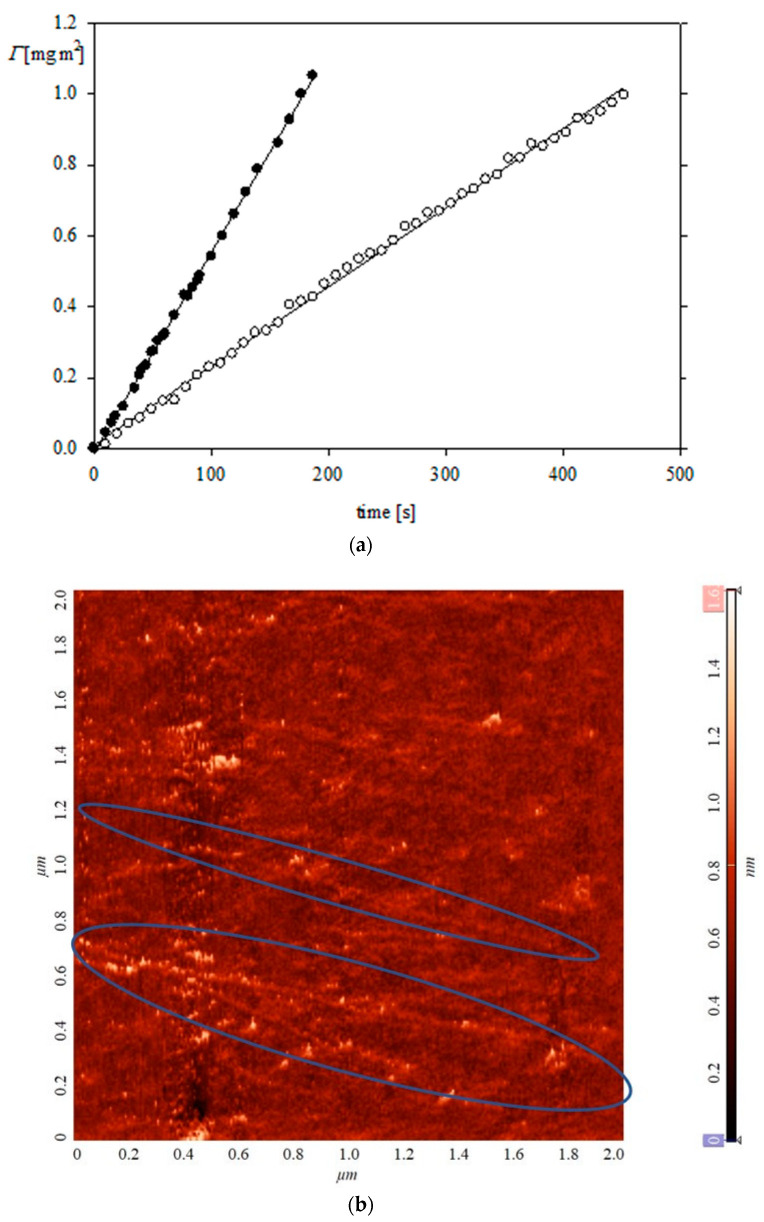
(**a**) The short-time kinetics of FGF 21 (empty points) and HSA (full points) molecule adsorption on the silica/PDADMAC substrate determined by OWLS. FGF 21 bulk protein concentration 2 mg L^−1^, flow rate 1.1 × 10^−3^ cm^3^ s^−1^, pH 7.4, 0.01 M NaCl. HSA bulk protein concentration 5 mg L^−1^, flow rate 2.5 × 10^−3^ cm^3^ s^−1^ (pH 7.4, 0.01 M NaCl). The solid lines show the linear fit characterized by the slope 2.2 × 10^−3^ mg m^−2^ s^−1^ (FGF 21) and 5.5 × 10^−3^ mg m^−2^ s^−1^ (HSA), (**b**) the AFM image of the FGF 21 molecule layer on PDADMAC covered silica, FGF 21 bulk protein concentration 0.2 mg L^−1^, adsorption time 10 min, scan area 2 × 2 μm. Long PDADMAC chains decorated by the protein molecules were marked by blue ellipses.

**Figure 10 biomolecules-13-01709-f010:**
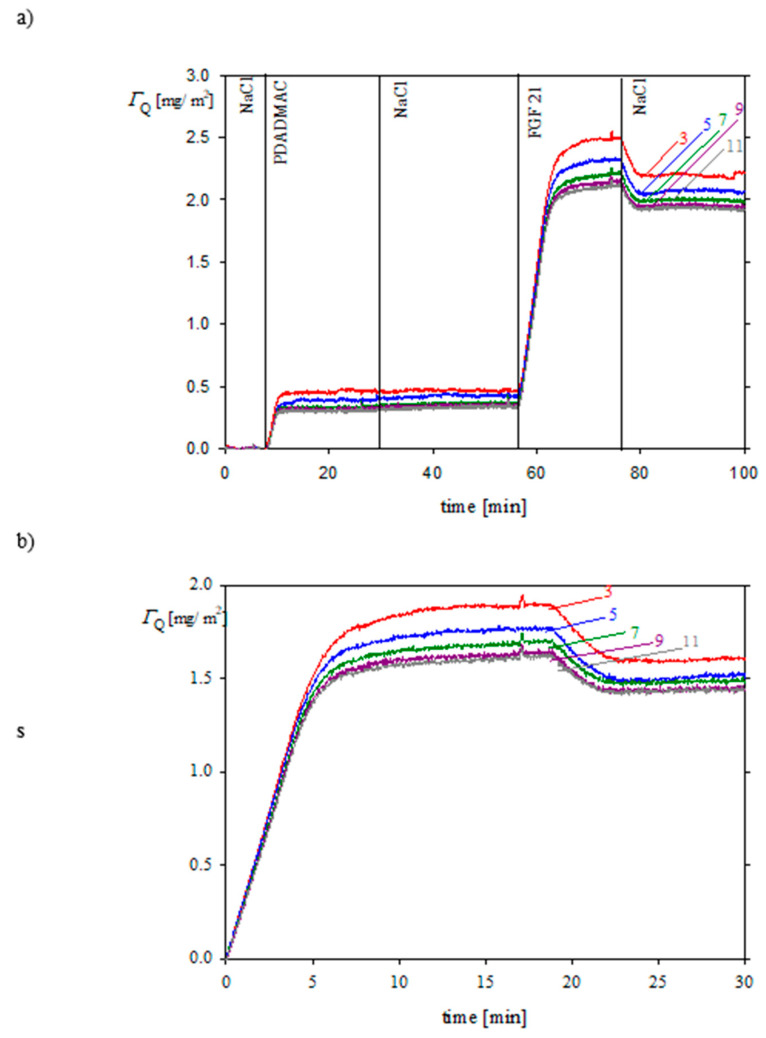
Kinetics of FGF 21 molecule adsorption on the silica/PDADMAC substrate determined by QCM (the protein coverage was calculated from the Sauerbrey model for various overtones, 3, 5, 7, 9, 11); flow rate 1.33 × 10^−3^ cm^3^ s^−1^, bulk protein concentration 2 mg L^−1^, pH 7.4, 0.01 M NaCl. Part (**a**) the entire kinetic run; part (**b**) the adsorption of FGF 21 alone.

**Figure 11 biomolecules-13-01709-f011:**
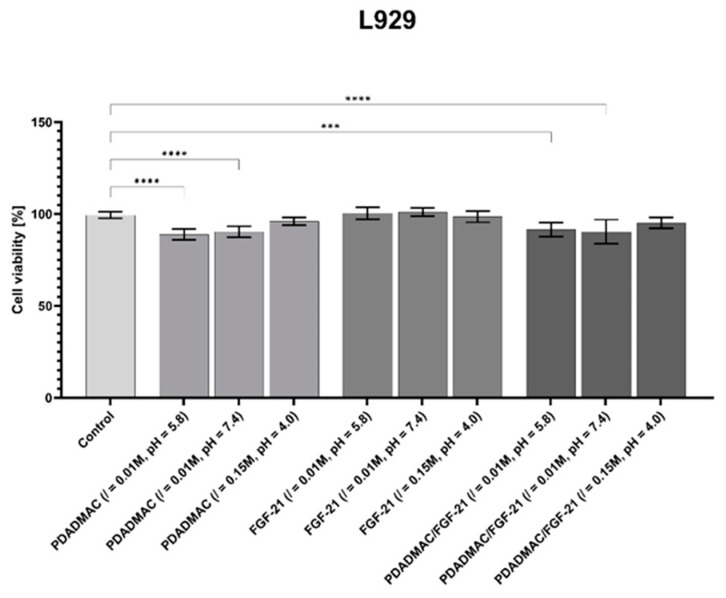
The influence of PDADMAC, FGF 21, and PDADMAC/FGF 21 coatings on the viability (in relation to untreated control) of L-929 cells. They were treated with components in different adsorption conditions (described in the picture as 0.01 M NaCl or 0.15 M NaCl, pH = 4.0, 5.8, or 7.4) and kept in regular cell culture conditions. After 24 h AlamarBlue™ reagent was added to cell culture according to the manufacturer’s instruction. The experiment was performed in three independent replicates. One-way ANOVA and Dunnett’s multiple comparisons test statistical analysis (GraphPad Prism) show significant differences. *** *p* < 0.001, **** *p* < 0.0001.

## Data Availability

The data are available on request.

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
