# Peer review of "Mechanisms of Fibroblast Growth Factor 21 Adsorption on Macroion Layers: Molecular Dynamics Modeling and Kinetic Measurements"

_biomolecules, 2023, doi:10.3390/biom13121709_

Round 1
Reviewer 1 Report
Comments and Suggestions for Authors
1. Major
There are comments about the author's title, "Mechanism of Fibroblast Growth Factor Adsorption on Macroion Layers: Molecular Dynamics Modeling and Kinetic Measurements". Although FGFs have structural similarities, their structures and behaviors are very diverse. The title presented by the author may lead to misunderstanding as it applies to all FGFs. Therefore, I hope that the author will change the title to "Fibroblast Growth Factor 21" more clearly.
2. Minor
1) The author would like to display the structural expression in Figure 1 in a ribbon or cartoon format to facilitate readers' understanding.
2) In Figure 3, please enlarge and show at least one of the dimer-aggregated forms of FGF21.
3) Regarding Figure 9B, it is difficult to understand the author's intention just by looking at the picture. I wish it would be shown a little more kindly to compare the state before and after bulk. At the very least, I hope that the “elongated macroion structures,” which the author said can be clearly seen, can be used to help readers understand them through auxiliary lines. In many cases, readers who lack understanding of AFM results often do not know what a protein is.
Reviewer 2 Report
Comments and Suggestions for Authors
Overall this was a well done study of the physicochemical solution properties and adsorption behavior of fibroblast growth factor 21 (FGF). Adsorption to silicon surfaces and silicon modified by preadsorption of the cationic polymer poly(diallyldimethylammonium chloride) (PDADMAC) was studied. The protein size and shape, cross-section area, the dependence of the nominal charge on pH and isoelectric point of 5.3 were acquired as well as the kinetics of its adsorption to silicon and PDADMAC preadsorbed silicon. The cationic polymer preadsorption increased the adsorption of FGF considerably, to a close packed array. The effect of FGF preadsorption to cell culture multiwell dishes on cell viability was also characterized using, CHO-K1/L-929 cells, showing modest increases in cell viability when FGF was preadsorbed to PDADMAC preadsorbed silicon in comparison to the PDADMAC preadsorbed silicon control.
Need to fix: The legends in several figures were not sufficient to identify the data in the body of the figure because the color used in the body was not shown in the figure legends i.e. the symbols in the legend were black but the symbols in body were various colors. I am referring to figures 4 and 5.
